# A Pathway to Precision Medicine for Aboriginal Australians: A Study Protocol

**DOI:** 10.3390/mps4020042

**Published:** 2021-06-21

**Authors:** Yeu-Yao Cheng, Jack Nunn, John Skinner, Boe Rambaldini, Tiffany Boughtwood, Tom Calma, Alex Brown, Cliff Meldrum, Marcel E. Dinger, Jennifer A. Byrne, Debbie McCowen, Jayden Potter, Kerry Faires, Sandra Cooper, Kylie Gwynne

**Affiliations:** 1Poche Centre for Indigenous Health, The University of Sydney, Camperdown, NSW 2050, Australia; john.skinner@sydney.edu.au (J.S.); boe.rambaldini@sydney.edu.au (B.R.); tom.calma@sydney.edu.au (T.C.); 2Faculty of Medicine and Health, The University of Sydney School of Medicine, Camperdown, NSW 2050, Australia; 3School of Psychology and Public Health, La Trobe University, Melbourne, VIC 3086, Australia; jack.nunn@latrobe.edu.au; 4Australian Genomics Health Alliance, Melbourne, VIC 3052, Australia; tiffany.boughtwood@mcri.edu.au; 5Murdoch Children’s Research Institute, Melbourne, VIC 3052, Australia; 6South Australian Health and Medical Research Institute, Adelaide, SA 5000, Australia; alex.brown@sahmri.com; 7NSW Health Pathology, Sydney, NSW 2065, Australia; cliff.meldrum1@health.nsw.gov.au; 8School of Biotechnology and Biomolecular Sciences, UNSW, Sydney, NSW 2052, Australia; m.dinger@unsw.edu.au; 9New South Wales Health Statewide Biobank, New South Wales Health Pathology, Camperdown, NSW 2050, Australia; jennifer.byrne@health.nsw.gov.au; 10School of Medical Sciences, Faulty of Medicine and Health, The University of Sydney, Camperdown, NSW 2006, Australia; 11Armajun Aboriginal Health Service, 1 Rivers Street, Inverell, NSW 2360, Australia; dmccowen@armajun.org.au (D.M.); potsy2009@gmail.com (J.P.); admin@armajun.org.au (K.F.); 12Kids Neuroscience Centre, Children’s Hospital at Westmead, Westmead, NSW 2145, Australia; sandra.cooper@sydney.edu.au; 13Discipline of Child and Adolescent Health, The University of Sydney, Camperdown, NSW 2006, Australia; 14The Children’s Medical Research Institute, Westmead, NSW 2145, Australia; 15Faculty of Medicine and Health Sciences, Macquarie University, Sydney, NSW 2113, Australia

**Keywords:** co-design, precision medicine, personalised medicine, Aboriginal health, participatory research, genomics, Aboriginal and Torres Strait Islanders, Australian

## Abstract

(1) Background: Genomic precision medicine (PM) utilises people’s genomic data to inform the delivery of preventive and therapeutic health care. PM has not been well-established for use with people of Aboriginal and Torres Strait Islander ancestry due to the paucity of genomic data from these communities. We report the development of a new protocol using co-design methods to enhance the potential use of PM for Aboriginal Australians. (2) Methods: This iterative qualitative study consists of five main phases. Phase-I will ensure appropriate governance of the project and establishment of a Project Advisory Committee. Following an initial consultation with the Aboriginal community, Phase-II will invite community members to participate in co-design workshops. In Phase-III, the Chief Investigators will participate in co-design workshops and document generated ideas. The notes shall be analysed thematically in Phase-IV with Aboriginal community representatives, and the summary will be disseminated to the communities. In Phase-V, we will evaluate the co-design process and adapt our protocol for the use in partnership with other communities. (3) Discussion: This study protocol represents a crucial first step to ensure that PM research is relevant and acceptable to Aboriginal Australians. Without fair access to PM, the gap in health outcome between Aboriginal and non-Aboriginal Australians will continue to widen.

## 1. Introduction

Genomic precision medicine (PM) leverages established understanding about genomic variation associated with wellness and disease to tailor medical treatment for an individual through targeted analyses of medically relevant information within an individual’s sequenced DNA. PM can positively impact all stages of clinical management, improving diagnosis, pinpointing appropriate preventive measures, and enabling therapeutic interventions [1,2]. To make PM more effective, genomic researchers group DNA sequence information from diverse global sub-populations, such as shared ancestry groupings, as people within these groupings will have a greater proportion of shared DNA traits [3]. These groups can affect the nature and direction of the research undertaken and the interventions that are available to those groups.

While PM is already being used worldwide to improve lives, its utility and effectiveness is not maximised for individuals with Aboriginal and Torres Strait Islander ancestry as there is less genomic data from people within these groupings [4,5]. There are no current population-specific data for Aboriginal Australians curated within the Genome Aggregation Database [6,7]. The deficit affects equity of access to PM for Aboriginal people, who represent a culturally and linguistically diverse set of communities at greater risks of poorer health outcomes [3,8]. In the era of rapidly advancing medical care and technology, there is a serious risk that this inequity will widen the already prominent health and life expectancy gaps between Aboriginal and non-Aboriginal populations [9,10]. The Australian government and state governments have developed health genomics plans and policy frameworks that include various strategies to increase the use of PM for Aboriginal Australians [11].

Genomic research is particularly sensitive and complex because genetics has been used in the past to justify and perpetuate racial discrimination [12,13,14]. Past research practice has positioned Indigenous peoples as research subjects rather than equal partners in the research process [15]. Historically, the lack of study transparency and research malpractice has also undermined trust in many scientific investigations and heightened concerns among Indigenous peoples about sharing personal health information [12]. While some people may have concerns regarding genetic research, such as genetic discrimination in employment, difficulty in obtaining insurance, and inappropriate use of genomic data in law enforcement (e.g., fabricated DNA evidence in crime scenes) [14], Indigenous peoples may have additional concerns specific to their cultural, social, and collective contexts. This may include allegations of genetic inferiority, threats to cultural beliefs, fear of exploitation for commercial purposes (e.g., drug development), and inappropriate use in Native Title claims or exclusion from government assistance [13]. These concerns highlight the importance of adopting a collaborative approach to conducting genomic research involving Indigenous peoples, one that partners with Indigenous communities and incorporates Indigenous perspectives and culture into study design and conduct.

Participatory Action Research (PAR) is an umbrella term for study models such as “co-creation” or “co-design”, which embraces a participatory philosophy [16]. Placing impacted communities at the heart of the process, the benefits of co-design include: the empowerment of vulnerable communities, better identification of priority topics relevant to the communities, generation of innovative ideas, reduction of scientific bias, and the potential for long-term collaboration on future projects [17,18]. In co-design, Aboriginal communities are involved in every step of the study: the development of the research question, design of the study methodology, determining governance of data, and the interpretation and dissemination of results. These stages of the study are continuously evaluated and refined to reflect the interest and values of the participating communities and co-design partners. “Involvement” is when power is shared with the research participants to actively contribute to the process of research [19] and is distinct from “engagement” where only information and knowledge is shared [20].

Crucially, the purpose of co-design is not limited to increasing the participation of Aboriginal people in genomic research but extends to improving research and service design as a target in its own right. Previous studies have shown that even when health services are available, desired health outcomes are not achieved if they are not designed for and with Aboriginal people [21]. Aboriginal health workforce development strategies (e.g., training of Aboriginal genetic counsellors and pathology collection professionals) should also be developed in parallel to deliver health care in a culturally safe manner. Aboriginal workforce training will not only support greater participation in, and benefits realised from, genomic PM—but those trained will also be able to meet other medical needs for their communities.

PM may play a role in closing the health gap for Aboriginal Australians, and will rely on co-design to enable the development and appropriateness of such health services [22,23]. In this paper, we propose a protocol for community involvement and co-design to collect aggregated genomic data.

## 2. Study Design

The current study is an iterative qualitative study that aims to begin the co-design process with three different Aboriginal communities. We bring together experts in five important areas: Aboriginal culture, community engagement, research co-design, genomics, and public policy. As shown in Figure 1 and described in greater detail later, the study consists of five main phases: Preparation, Community Involvement, Data Collection, Data Analysis and Dissemination of Results, and Adaptation.

While the overall study structure is similar for all communities involved, the specific details of the study within each respective community could vary considerably owing to the co-design process. For example, there may be changes to the language, images, and analogies used to explain PM, and there may be different arrangements with respect to management and storage of samples and data. Furthermore, during the Adaptation phase, ideas generated with the first community will help shape future co-design workshops with subsequent communities, highlighting the strength and iterative nature of Participatory Action Research. It is predicted that the process of adapting the protocol will take three months for each new community.

## 3. Methods

### 3.1. Phase I: Preparation

#### 3.1.1. Ensuring Aboriginal Governance

Previous studies have demonstrated that research with Aboriginal people that is led by Aboriginal researchers is more culturally relevant, effective, and readily adopted [24]. Our core research team of 10 researchers includes five Aboriginal Chief Investigators (CI) who have substantial research experience. The CI’s will be responsible for the overarching guidance, design, and conduct of the entire project.

In addition to the core research team, an independent Project Advisory Committee (PAC) [25] will be established and jointly chaired by an Aboriginal Elder nominated by the community and the director of the Poche Centre for Indigenous Health of the University of Sydney who is an Aboriginal Elder and senior researcher. Each community will have its own PAC and will meet bi-monthly. The primary role of the PAC will be to ensure Aboriginal control, governance, and timely completion of the project. The PAC will oversee the cultural safety and integrity of the project, as well as advise on the dissemination and sharing of results to influence policy and practice based on research findings. An annual progress report will be provided to the Aboriginal Health and Medical Research Council (AHMRC) Ethics Committee, which reviews all presentations and publications prior to presentation and submission. The chairs of the different PACs will have the opportunity to communicate, which allows for the consideration of feedback and experiences from communities other than their own.

#### 3.1.2. Developing Research Team Cultural Competency

There are currently no specific Australian guidelines for cultural competency training, requirements, or standards for researchers involved in Aboriginal genetic studies [26]. We will follow the guidelines from National Health and Medical Research Council (NHMRC) [27] and strategies outlined in Genomic Partnerships [28] to engage and partner with Aboriginal peoples for genomic research. All investigators will either have extensive experience working with Aboriginal communities or will complete cultural competence training [29] and pre-readings on selected Aboriginal genetic research articles. Investigators will also familiarize themselves with the recommendations of the Aboriginal Knowledge and Intellectual Property Protocol [30]. All investigators will be expected to interact with Aboriginal communities with respect, humility, empathy, and willingness to listen throughout all the stages of the study. Feedback about Investigators and their behaviours when working with Aboriginal people and communities will be managed through each PAC.

#### 3.1.3. Initial Community Consultation to Explore Views of Impacted Communities

In partnership with Armajun Aboriginal Health Service (AHS), the Poche Centre has developed a plain English summary of the PM project. A video created by the National Centre for Indigenous Genomics at the Australia National University [31] has been adapted to explain genomic precision medicine visually and in plain English. Armajun AHS proposed an Aboriginal community in New South Wales (NSW) for initial community involvement because they had some experience and health literacy with genetic medicine. In addition, our research team had been collaborating with Armajun AHS for more than five years on several co-design research projects in oral health, cardiovascular disease, and lung health. This research was valued by the community and the research team. The introductory video [32] and information sheets were made widely available in the community. An online survey [33] was conducted by the Armajun AHS and the Local Aboriginal Land Council (LALC) to determine community support for the project. An informal discussion was used to explore people’s views and preferences for involvement. Members of the community also helped to improve the dissemination plan for letting more people in the community know about the planned study.

An interviewer assisted online anonymous survey was conducted via SurveyMonkey (Appendix A) [34] by the Aboriginal Community Controlled Health Services (ACCHS) and the LALC. The survey consisted of eight questions gathering the individual’s basic demographic information (age, Indigenous status, geographical location), thoughts on the PM video, and support (or the lack thereof) for the study. Through the consultation, it was agreed that at least 10% Aboriginal community support was required for the LALC and Armajun AHS to provide letters of support for the human research ethics application. The survey indicated a high level of support and willingness to co-design, with 100% (n = 51) community members surveyed supporting the study in less than three weeks. Of the 51 participants who watched the video and responded to the survey (47% male, 78% younger than 55 y), 19 (42%) shared their thoughts on the project. Participants suggested face-to-face events as the preferred method for co-design. One-hundred percent (n = 51) of respondents believed it was a good idea to collect DNA samples from the community to form the DNA library. Letters of support were subsequently provided from the ACCHS and the LALC (Appendix A).

### 3.2. Phase II: Community Involvement

#### 3.2.1. Selection of Aboriginal Communities

Where consent is granted by the local AHS following the Preparation phase, we will invite prospective communities to participate in co-design. We will follow the actions of the dissemination plan developed with community members during the “initial community consultation” of the Preparation phase. This includes engaging in mainstream print, TV media, social platforms, and advertisements in public places such as shop windows. The co-design workshops will be open to all members of the communities. Local AHSs will also be asked to nominate suitable prospective participants. Where possible, we will ensure that the group is heterogenous as we seek diverse experiences and opinions related to genetic testing.

#### 3.2.2. Inclusion and Exclusion Criteria

People who identify as Aboriginal aged 18 years and above will be eligible to participate in the co-design workshop that will take place at the LALC and ACCHS.

#### 3.2.3. Consent to Co-Design

Each participant will be provided a written participant information statement and consent form developed by the AHMRC (Appendix A) [35]. Participants will be reminded that their participation is entirely voluntary, information gathered during surveys and discussions are anonymous, and withdrawal from the study is allowed at any point without any consequences or implications on their future access to healthcare.

### 3.3. Phase III: Data Collection

#### 3.3.1. Structure of the Co-Design Workshop

An Aboriginal Elder will open the workshop with local customs and traditions specific to their community. The opening will be followed by an introduction by the director of the Poche Centre for Indigenous Health about the research institution and staff members present. A 4-min introductory video explaining PM [32] will be played and plain English information sheets will be handed to attendees. The video begins with the Acknowledgement of Country, followed by the explanation of the concept of DNA. The video uses analogies, which illustrate DNA as stories in the blood written in languages, that can be understood by clinicians and scientists to discover why one person gets sick. The video describes DNA as “the story is passed down from our ancestors and that people can share stories with others in the same geographical area even if they are not from the same family.” The video highlights that the DNA stories allow doctors to understand why some communities are more prone to certain genetic disorders or why certain treatment might work in some family members but not others. Finally, the video states the aim of the project: to create a baseline DNA story so that Aboriginal people can access PM.

Participants will be encouraged to ask questions and provide thoughts on the video and the study. The CIs present will ask open-ended questions, including exploring priority topics relevant to precision medicine. Apart from helping to spark ideas and discussion, the CI’s will also be required to ensure an optimal dynamic for the exchange of opinion. The CI’s will follow the verbal and non-verbal cues of senior Aboriginal participants in the discussion [36]. The CI’s will routinely ask questions such as “is this what you had in mind?” to confirm that the interpretation of the idea is congruent with that of the participants. The “NSW Health Statewide Biobank Consent Toolkit” [37] and the “Patient Information Booklet: Genetic and Genomic Testing” [38] provide standards and guidance for biobanking and consent procedures. We will review these documents along with those from other states to ensure key themes related to PM are co-designed with communities during the workshop. Areas requiring co-design include participant recruitment, consenting, DNA sample collection and storage, data governance and sovereignty, and the dissemination of results and incidental findings. Additionally, workforce involvement (training of local Aboriginal people as genetic counsellors and pathology collectors) will be discussed with the participants. This leverages previous work by the research team, which identified the urgent need to address the insufficient supply of Aboriginal health professionals [39], and Poche Centre’s ongoing mission to build and support education and career pathways for Aboriginal people [40].

Although participants will be encouraged during the workshop to voice their thoughts or ask questions, we respect that not all participants are comfortable sharing ideas in front of the larger group. Smaller group discussions will also be facilitated, with the groups reporting feedback both verbally and in writing. Alternative methods of providing feedback and asking questions will be provided during the meeting, including written notes, which can be submitted anonymously if the participant chooses. Regular breaks will ensure that participants can approach members of the research team to discuss ideas or concerns privately if preferred. At the end of the workshop, contact information for the research team will be provided to the participants.

#### 3.3.2. Documentation the Co-Design Process

Discussions during the co-design workshop will be summarised with pseudonyms assigned to participants. The note taker will verbally repeat the notes taken and check with participants for accuracy and to ask if they wish to add anything. Locations, communities, and participants will be de-identified to protect confidentiality and privacy. In addition, notes will be shared after the event, with participants given an opportunity to provide further feedback on the notes in writing. The co-design process will also be reported using the STARDIT reporting tool, with a report co-created with the community [41]. The report will contain data about who was involved, how, and in which tasks. Information about the co-design method, the data analysis methods, reported enablers and barriers for involvement, and any other impacts and outcomes from the co-design process will also be reported.

### 3.4. Phase IV: Data Analysis & Dissemination of Results

#### 3.4.1. Data Analysis

Quantitative data gathered through surveys regarding the group’s demographic makeup will be analysed through descriptive statistics. For the qualitative data, we will perform an inductive thematic analysis using NVivo to organize participant’s responses into key themes. Coding and thematic analysis of qualitative data will be carried out by two members of the study team and checked by another author, following best practices for enhancing validity in qualitative methods [42,43]. At least one of the three investigators involved in coding and analysis will be Aboriginal. The core research team will meet to review the findings and identify outstanding or representative quotes for future presentation of the results. Preliminary findings will be discussed with each PAC.

#### 3.4.2. Returning Results to the Community

Results of the co-design study and a draft protocol for the PM study will be disseminated into the local Aboriginal community through community meetings and printed research summaries (including Plain English summaries). We will work in collaboration with ACCHS to hold community meetings and information evenings, which both participants and non-participants of the study can attend. Community members will be encouraged to provide feedback and comments on the draft local protocol and modified introductory video and information sheet to ensure that our interpretation is consistent with the views of the community and co-design participants.

Results will be published in peer reviewed journals and presented at professional conferences. ACCHSs participating in the study will be invited to contribute to these publications and presentations. We will acknowledge the sources of information and those who have contributed to the research through authorship and acknowledgement in resulting publications, meetings with community members and conference presentations. We will also acknowledge the cultural property rights of Aboriginal peoples in relation to knowledge, ideas, cultural expressions, and cultural materials by including AHS representatives as Chief Investigators on the study.

#### 3.4.3. Dissemination of Results

An integrated knowledge-to-action (KTA) plan will be developed to detail our engagement of key stakeholders in the research process and disseminate ongoing findings. The types of KTA activities will be determined in collaboration with our project partners, co-design participants, and the PAC. The diversity of stakeholders and organisations involved our project will ensure that our research findings inform policy and practice, leading to the development of personalised medicine. Our KTA strategies will include multiple strategies such as social media (e.g., Facebook, Twitter, Instagram, and LinkedIn), mass media communications, policy papers, conference presentations and open access journal publications, short video and podcasts, and public events (e.g., barbecues, picnics, and campfires).

### 3.5. Phase V: Adaptation

#### 3.5.1. Transference of Data from the Co-Design Team to the Precision Medicine Team

The PM project consists of three stages: the co-design study, workforce involvement, and biobanking. The co-design study serves to create a framework and inform the subsequent stages of the PM project. All investigators will have access to findings of the co-design study (e.g., stakeholder’s view on appropriate biobanking and consenting). Further details on data management of the latter stages of the PM project (e.g., DNA samples, DNA sequences) will be co-designed with the communities. The processes of data transference used in the project will be consistent with the principles of Participatory Action Research where stakeholders collectively decide upon roles, responsibilities, and data access [23]. 

#### 3.5.2. Revising and Refining the Precision Medicine Project

Appendix A is a plain English summary of the most up-to-date version of the PM project, which will be revised based on findings of the current participatory study. Following the conclusion of this study, investigators will meet to discuss how to refine the PM project with input from the PAC. Elements of the PM project include the following: the consent process, collection and storage of samples, governance of data, and reporting findings will be reconstructed to align with the views of participants and stakeholders.

#### 3.5.3. Reporting and Evaluating Co-Design

To date, there are no guidelines on how to objectively assess the quality of co-design processes. However, New Philanthropy Capital (NPC) have suggested that co-design can be assessed based on three areas: (1) benefits to the participants, (2) the quality of the process, and (3) the insights obtained to achieve the intended outcome [44]. Although the discussion of an objective evaluation of co-design is beyond the scope of this paper, the authors appreciate the importance, challenges, and potential ways of evaluating co-design, and they will be selectively using a combination of resources to appraise and uphold the integrity and quality of our work. We propose the use of Standardised Data on Initiatives (STARDIT) [41] to report, evaluate, and keep public records of every stage of the co-design process.

## 4. Discussion

To make precision medicine available to Aboriginal and Torres Strait Islanders, the broader PM project uses the Five Aim Approach (Figure 2): (1) Protocol co-design, (2) Capacity building (training of Aboriginal professionals for genetic counselling and pathology collection), (3) Culturally safe sample collection, (4) Data analysis & bioinformatics, and (5) Translation. The manuscript is the protocol paper for the first part (Protocol co-design) of the broader PM study, with a focus on creating a methodology for community engagement and involvement that is feasible and acceptable. It represents a critical first step to improve and enable PM related health service delivery and to close the health gap between Indigenous and non-Indigenous populations. While co-design is the first part of PM project, co-production is continuous throughout the entire PM project, with Aboriginal communities, CIs, and the PAC involved as co-researchers for all the stages of the project.

The value of Participatory Action Research with Indigenous communities is well documented [17,18,22,23]. The concept of “involvement” gives rise to the main strengths of the study: fluidity, adaptability, and cultural safety. As the nature of our topic (i.e., genomics research and research with Aboriginal communities) is highly complex and sensitive, co-design becomes not only preferred but necessary due to its emphasis on both the process and result of research. Although time consuming at the initial stages, by involving participants early on, we ensure that the study is feasible, the topic areas are relevant and meet stakeholder needs, and the study process upholds the required cultural respect and integrity. In the long-run, using a co-design process saves time, promotes the usage of health services, and elicits superior health outcomes [21]. Other benefits specific to our PM project that support the use of co-design include the building of strong and committed community partners and enhancing skills and knowledge in the Aboriginal community about genomics and health. 

By prioritising equity and valuing culture and community over research outputs, the current paper delivers the necessary guiding framework (culturally acceptable recruitment of participants, informed consent, and governance of data and samples) to answer important questions about data ethics, security, and quality associated with genomic research. We bring together experts from various disciplines to collaboratively develop the protocol according to published guidelines on ethical engagement of Aboriginal people in genomic research [18] and previously identified enablers for effective health service delivery for Aboriginal Australians [21]. The next step for the study is to apply for ethics approval from the AHMRC, after which we will commence with community involvement.

### Limitations

The protocol shares limitations that are present in many other Participatory Action Research studies. This includes not being able to elucidate the precise details (e.g., content of the co-design workshop) at the time of writing this study protocol and the lack of standardised ways to report co-design. Despite these limitations, a major benefit of co-design is the discovery of relevant topics, enablers, and barriers that are not yet known to the research team. At this point, we provide a general direction for co-design topics that is relevant to all genomic studies, which will be refined by participants. We will transparently document and evaluate the co-design process.

## Figures and Tables

**Figure 1 mps-04-00042-f001:**
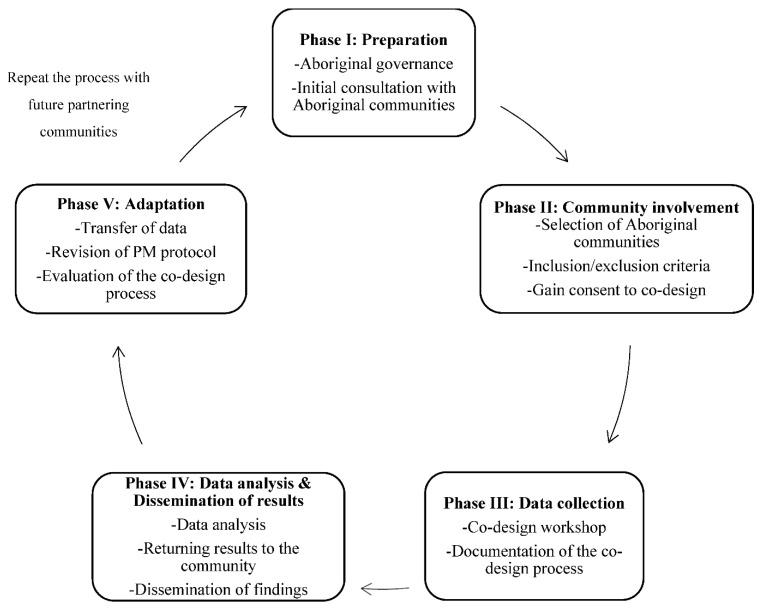
Five Phases of the Co-design Protocol.

**Figure 2 mps-04-00042-f002:**
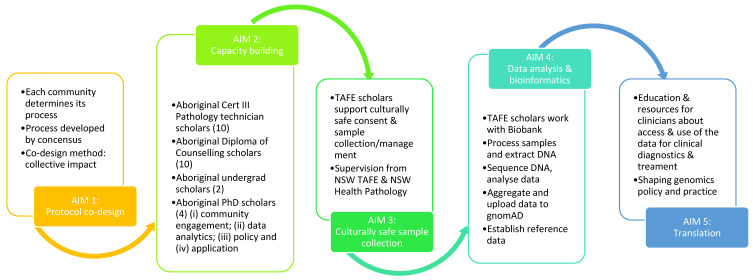
Five aim approach to enable Aboriginal Precision Medicine.

## Data Availability

The datasets used during the current study are available from the corresponding author on reasonable request.

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
