# Peer review of "A Pathway to Precision Medicine for Aboriginal Australians: A Study Protocol"

_mps, 2021, doi:10.3390/mps4020042_

Round 1

Reviewer 1 Report

Manuscript ID: mps-1250372
Type of manuscript: Protocol
Title: A Pathway to Precision Medicine for Aboriginal Australians: A Study 
Protocol. Submitted to section: Public Health Research.

The manuscript is a qualitative study protocol using a co-design method to establish a biobank of DNA samples from 3 Aboriginal Australian communities in NSW to inform precision medicine. It is a thoughtful and well-designed protocol in which the research comprises 50% as Aboriginal Chief Investigators with “substantial research experience” but it was not apparent whether they had genomics research experience. [Q-Question to be responded to]].

The whole aim of such a biobank is to “..improve diagnosis”, enhance preventative strategies and optimise therapeutic interventions- presumably that includes quality use of medicines. Some examples would have enhanced the manuscript as although much is written on precision medicine and genomics, there are few useful examples, with the exception of medicines.[Q]. A particular strength of the co-design is Capacity Building (Aim 2) whi ch is rightfully embedded in the design.

Other specific comments are:

  1. It was unclear whether just demographic data will be collected as this could be a missed opportunity. The very successful UK biobank collects data on comorbidities, comedications and overall health (e.g. sleep). Consideration ought to be given to including such data.[Q]
  2. Phase II: although the following may appear tangential to the manuscript, they are important to be included in the manuscript as other researchers of First Nations People genomics can be informed as an example of a well planned and collaborative co-design pathway.
  3. It was not clear whether in family members are to be included with the potential for biasing the data[Q].
  4. In addition, Phase II ought to make it clear that participants can be given access to their personal genomics results and their interpretation [Q].
  5. It was not stated in what biosamples will be collected. The supplementary file indicated blood and saliva. It was not explained why both are required and especially blood which has specific meaning and relevance to Aboriginal culture and practice[Q].
  6. There was no mention that the biobank can be accessed by other researchers, not the current 10 chief investigators [Q].
  7. No mention that participants can withdraw consent and have their sample destroyed[Q]

Minor Comments.

  1. References: ref 4 makes no mention of Aboriginal and Torres Strait Islander ancestry etc; ref 6 missing volume, page numbers; ref 9 has no journal or volume; ref 10, 11, 15, 25 require the URL; ref 19, 20, 27, 28, 31, 35,37,38 need to be adequately referenced for readers of the paper.
  2. The Supplementary material p 10 “ This is an example of what the story of the DNA in the International DNA Library will look like. Although this is correct, only a genomics researcher will be able to read and understand it.
  3. Will RNA be collected?

Author Response

Dear Reviewer,

We would like to thank you for taking the time to review our manuscript and provide valuable feedback. 

Please see the attachment for detailed responses to your questions.

Reviewer 2 Report

Cheng et al. propose a pathway for precision medicine in Aboriginal Australians.

This is a timely idea.

Specific comments include:

  1. Lines 131, 133, 134 etc.: why “will” and not was?
  2. And then why is line 158 “has”, but then line 160 “were” etc.?
  3. Line 161: why genetic medicine, while the title and the manuscript is mostly on precision medicine?
  4. Line 173: how many people were contacted, how many responded etc.? If this is a proposal: what numbers are aimed for?
  5. Line 378: what new tools?
  6. There are also minor issues with inconsistent capitalization etc.

Author Response

(The authors gave the same response as above.)

Round 2

Reviewer 2 Report

The authors have made some changes to improve the manuscript. Numbers remain low.

Author Response

Dear Reviewer, 

Thank you for comments. We have thoroughly gone through the manuscript again and made revision to ensure that our language and grammar are consistent. Please see the attachment/

Regarding your comments on the study design, at this time we proposed a protocol to collaborate with the participating Aboriginal communities to develop an approach to collect and store genetic samples in a culturally safe manner. The second part (i.e. biosampling) of this two-part process is not the focus of the current manuscript and will be addressed in a separate paper once available. We look forward to sharing the biosampling protocol paper with the scientific community in the near future. 

Best regards,

Kevin